# Uniform Treatment of Integral Majorization Inequalities with Applications to Hermite-Hadamard-Fejér-Type Inequalities and *f*-Divergences

**DOI:** 10.3390/e25060954

**Published:** 2023-06-19

**Authors:** László Horváth

**Affiliations:** Department of Mathematics, University of Pannonia, Egyetem u. 10., 8200 Veszprém, Hungary; horvath.laszlo@mik.uni-pannon.hu

**Keywords:** majorization inequalities, convex functions, signed measures, Hermite-Hadamard-Fejér-type inequalities, refinement, *f*-divergences, 26D15, 26A51, 94A17

## Abstract

In this paper, we present a general framework that provides a comprehensive and uniform treatment of integral majorization inequalities for convex functions and finite signed measures. Along with new results, we present unified and simple proofs of classical statements. To apply our results, we deal with Hermite-Hadamard-Fejér-type inequalities and their refinements. We present a general method to refine both sides of Hermite-Hadamard-Fejér-type inequalities. The results of many papers on the refinement of the Hermite-Hadamard inequality, whose proofs are based on different ideas, can be treated in a uniform way by this method. Finally, we establish a necessary and sufficient condition for when a fundamental inequality of *f*-divergences can be refined by another *f*-divergence.

## 1. Introduction

The theory of majorization is a useful mathematical tool, and many important and interesting inequalities can be obtained by combining it with the theory of convex functions. The basic concepts of majorization include the following binary relations for finite sequences of real numbers:

**Definition** **1.**
*Let $x:=\left(x_{1},\ldots ,x_{n}\right)\in R^{n}$ and $y:=\left(y_{1},\ldots ,y_{n}\right)\in R^{n}$.*

*(a) We say that x is weakly majorized by y, denoted as $x{≺}_{w}y$, if*

(1)
$$\sum_{i=1}^{k}x_{\left[i\right]}\leq \sum_{i=1}^{k}y_{\left[i\right]},\;k=1,\ldots ,n,$$

*where $x_{\left[1\right]}\geq x_{\left[2\right]}\geq \ldots \geq x_{\left[n\right]}$ and $y_{\left[1\right]}\geq y_{\left[2\right]}\geq \ldots \geq y_{\left[n\right]}$ are the entries of x and y, respectively, in decreasing order.*

*(b) We say that x is majorized by y, denoted as x≺y, if (Equation 1) holds, and in addition,*

$$\sum_{i=1}^{n}x_{\left[i\right]}=\sum_{i=1}^{n}y_{\left[i\right]}.$$



The fundamental inequality relating majorization and convexity is the Hardy–Littlewood–Pólya inequality, (see [1]).

**Theorem** **1.**
*Let C⊂R be an interval, let f:C→R be a convex function, and let $x:=\left(x_{1},\ldots ,x_{n}\right)\in C^{n}$ and $y:=\left(y_{1},\ldots ,y_{n}\right)\in C^{n}$.*

*(a) If x≺y, then*

(2)
$$\sum_{i=1}^{n}f\left(x_{i}\right)\leq \sum_{i=1}^{n}f\left(y_{i}\right).$$


*(b) If f is increasing and $x{≺}_{w}y$, then (Equation 2) also holds.*


Among the weighted versions of the previous result, we highlight the following inequality by Fuchs [2].

**Theorem** **2.**
*Let C⊂R be an interval, and let f:C→R be a convex function. If $\left(x_{1},\ldots ,x_{n}\right)\in C^{n}$, $\left(y_{1},\ldots ,y_{n}\right)\in C^{n}$ and $q_{1},\ldots ,q_{n}$ are real numbers, such that*

*(a) $x_{1}\geq \ldots \geq x_{n}$ and $y_{1}\geq \ldots \geq y_{n}$,*

*(b) $\sum_{i=1}^{k}q_{i}x_{i}\leq \sum_{i=1}^{k}q_{i}y_{i}$ $\left(k=1,\ldots ,n-1\right)$,*

*(c) $\sum_{i=1}^{n}q_{i}x_{i}=\sum_{i=1}^{n}q_{i}y_{i}$,*

*then*

$$\sum_{i=1}^{n}q_{i}f\left(x_{i}\right)\leq \sum_{i=1}^{n}q_{i}f\left(y_{i}\right).$$



The notion of majorization can be extended to the continuous case.

**Definition** **2.**
*Let φ, $\psi :\left[a,b\right]\to R$ be decreasing functions. We say that φ is majorized by ψ in symbols φ≺ψ if*

$$\int_{a}^{x}\varphi \left(t\right)dt\leq \int_{a}^{x}\psi \left(t\right)dt,\;x\in \left[a,b\right]$$

*and*

$$\int_{a}^{b}\varphi \left(t\right)dt=\int_{a}^{b}\psi \left(t\right)dt.$$



The next result is the integral version of the Hardy–Littlewood–Pólya inequality (see [3]).

**Theorem** **3.**
*Let φ, $\psi :\left[a,b\right]\to C$ represent decreasing functions, where C⊂R is an interval. Then, φ is majorized by ψ if and only if*

$$\int_{a}^{b}f\left(\varphi \left(t\right)\right)dt\leq \int_{a}^{b}f\left(\psi \left(t\right)\right)dt$$

*holds for every continuous and convex function f on C, such that the integrals exist.*


In the results related to the previous statement (majorization-type inequalities for integrals, see, e.g., the papers [4,5,6,7]), the conditions on the convex function are generally the same; it is defined on a compact interval and it is continuous. The proofs are usually based on different methods; the pointwise approximation of convex functions by smooth convex functions is a frequently used technique. Definition 2 can be naturally generalized by using measures and even signed measures, so Theorem 3 has extensions in these directions; see, e.g., the papers [7,8]. In this paper, we provide a general framework that offers a comprehensive and uniform treatment of the problem by providing conditions for the inequality
(3)$$\int_{\left[a,b\right]}f\circ \varphi d\mu \leq \int_{\left[a,b\right]}f\circ \psi d\nu ,$$
to be valid, where μ and ν are finite signed measures on a σ-algebra containing the Borel sets of $\left[a,b\right]$, and *f* is a convex function defined on an interval C⊂R. We obtain previously known results and solve this problem in new cases. We emphasize that neither the compactness of interval *C* nor the continuity of function *f* is required. The proofs only use the approximability of convex functions by piecewise linear convex functions (no smoothness condition is used). This result is well known when *C* is a compact interval (see [1]). We extend this statement to convex functions defined on arbitrary intervals, and show that the approximating sequence can always be chosen to be an increasing sequence. By using this, necessary and sufficient conditions are given for the inequality (Equation 3) to be fulfilled. As a consequence, some majorization-type inequalities for integrals are obtained. To apply these results, we deal with Hermite-Hadamard-Fejér-type inequalities and their refinements. Along with new results, we obtain unified and simple proofs of classical statements of Fink [9] and Florea and Niculescu [10]. We present a general method to refine both sides of Hermite-Hadamard-Fejér-type inequalities. The results of many papers on the refinement of the Hermite-Hadamard inequality, whose proofs are based on different ideas, can be treated in a uniform way by this method. Finally, we establish a necessary and sufficient condition for when a fundamental inequality of *f*-divergences can be refined by another *f*-divergence.

## 2. Preliminary Results

Positive and negative parts of a real number *x* are denoted by $x^{+}$ and $x^{-}$, respectively.

The complement of a set A⊂B, with respect to *B*, is denoted by $A^{c}$.

The σ-algebra of Borel sets and the σ-algebra of Lebesgue measurable sets on an interval C⊂R are denoted by $B_{C}$ and $L_{C}$, respectively.

Let $\left(X,A\right)$ be a measurable space. The unit mass at x∈X (the Dirac measure at *x*) is denoted by $\varepsilon_{x}$. Let μ be a signed measure on A. The total variation of μ is denoted by $\left|\mu \right|$. The real vector space of μ-integrable real functions on *X* is denoted by $L\left(X,\mu \right)$.

Let C⊂R be an interval with a nonempty interior. The following notations are introduced for some special functions defined on *C*:$$id_{C}\left(x\right):=x,\;p_{C,w}\left(x\right):={\left(x-w\right)}^{+},\;n_{C,w}\left(x\right):={\left(x-w\right)}^{-}\;x,w\in C.$$

We begin with two preparatory lemmas, which are important for what follows and are of interest in their own right.

**Lemma** **1.**
*Let $\left[a,b\right]\subset R$ with a<b, and let $\left(\left[a,b\right],A\right)$ be a measurable space, such that $B_{\left[a,b\right]}\subset A$ and μ is a finite signed measure on A. Assume φ, $\psi \in L\left(\left[a,b\right],\mu \right)$.*

*(a) If*

(4)
$$\int_{\left[a,x\right]}\varphi d\mu \leq \int_{\left[a,x\right]}\psi d\mu ,\;x\in \left[a,b\right],$$

*then*

$$\int_{\left[a,x\right]}\varphi d\mu \leq \int_{\left[a,x\right]}\psi d\mu ,\;x\in \left[a,b\right].$$


*(b) If (Equation 4) holds, and $\alpha :\left[a,b\right]\to R$ is a nonnegative and decreasing function, then*

$$\int_{\left[a,b\right]}\alpha \varphi d\mu \leq \int_{\left[a,b\right]}\alpha \psi d\mu .$$



**Proof.** (a) It can be assumed that $x\in \left[a,b\right]$. Choose a strictly increasing sequence ${\left(x_{n}\right)}_{n=1}^{\infty }$ in $\left[a,x\right]$, such that $x_{n}\to x$.Since both set functions
$$A\to \int_{A}\varphi d\mu \;\mathrm{and}\;A\to \int_{A}\psi d\mu ,\;A\in A$$
are (finite) signed measures on A, and ${\left(\left[a,x_{n}\right]\right)}_{n=1}^{\infty }$ is an increasing sequence converging to $\left[a,x\right]$, inequality (Equation 4) implies that
$$\int_{\left[a,x\right]}\varphi d\mu =\lim_{n\to \infty }\int_{\left[a,x_{n}\right]}\varphi d\mu \leq \lim_{n\to \infty }\int_{\left[a,x_{n}\right]}\psi d\mu =\int_{\left[a,x\right]}\psi d\mu .$$(b) Since α is decreasing on the compact interval $\left[a,b\right]$, it is Borel-measurable and bounded. According to $B_{\left[a,b\right]}\subset A$, this implies that αφ and αψ are also μ-integrable.We first assume that α is a simple decreasing function of the form
(5)$$\alpha =\sum_{i=1}^{k}c_{i}\chi_{I_{i}},$$
where
(6)$$c_{1}>\ldots >c_{k}\geq 0,$$
$I_{1},\ldots I_{k}$ are pairwise disjoint and nonempty intervals with $\overset{k}{\underset{i=1}{⋃}}I_{i}=\left[a,b\right]$ (these intervals can include open, closed, half-open intervals, and singletons; the upper endpoint of $I_{i}$ is the same as the lower endpoint of $I_{i+1}$
$\left(i=1,\ldots ,k-1\right)$), and $\chi_{I_{i}}$
$\left(i=1,\ldots k\right)$ denotes the characteristic function of $I_{i}$ with domain $\left[a,b\right]$. We introduce the intervals
$$J_{0}:=Ø,\;J_{i}:=\overset{i}{\underset{l=1}{⋃}}I_{l},\;i=1,\ldots ,k.$$By using (Equation 4), part (a), and (Equation 6), we obtain
$$\int_{\left[a,b\right]}\alpha \psi d\mu -\int_{\left[a,b\right]}\alpha \varphi d\mu =\sum_{i=1}^{k}c_{i}\int_{I_{i}}\left(\psi -\varphi \right)d\mu =\sum_{i=1}^{k}c_{i}\int_{J_{i}\setminus J_{i-1}}\left(\psi -\varphi \right)d\mu$$
$$=\sum_{i=1}^{k-1}\left(c_{i}-c_{i+1}\right)\int_{J_{i}}\left(\psi -\varphi \right)d\mu +c_{k}\int_{J_{k}}\left(\psi -\varphi \right)d\mu \geq 0.$$The general case follows from this and from the well-known result that there exists a sequence $\left(\alpha_{n}\right)$ of nonnegative and decreasing functions, such that each $\alpha_{n}$ has the same structure as (Equation 5) and $\alpha_{n}\to \alpha$, uniformly, on $\left[a,b\right]$.The proof is complete. □

We proceed with a simple but essential statement.

**Lemma** **2.**
*Let $\left[a,b\right]\subset R$ with a<b, and let $\left(\left[a,b\right],A\right)$ be a measurable space, such that $B_{\left[a,b\right]}\subset A$ and μ, ν are finite signed measures on A with $\mu \left(\left[a,b\right]\right)=\nu \left(\left[a,b\right]\right)$. Let $\varphi \in L\left(\left[a,b\right],\mu \right)$ and $\psi \in L\left(\left[a,b\right],\nu \right)$, such that $\int_{\left[a,b\right]}\varphi d\mu =\int_{\left[a,b\right]}\psi d\nu$. Then, for every w∈R, the following two assertions are equivalent.*

*(a)*

(7)
$$\int_{\left[a,b\right]}p_{R,w}\circ \varphi d\mu \leq \int_{\left[a,b\right]}p_{R,w}\circ \psi d\nu .$$


*(b)*

$$\int_{\left[a,b\right]}n_{R,w}\circ \varphi d\mu \leq \int_{\left[a,b\right]}n_{R,w}\circ \psi d\nu .$$



**Proof.** We only prove that (b) follows from (a); the converse statement can be handled similarly. By introducing the sets (these sets may be empty, and they belong to A)
(8)$$A_{\varphi }:=\left\{t\in \left[a,b\right]|\varphi \left(t\right)\geq w\right\},\;A_{\psi }:=\left\{t\in \left[a,b\right]|\psi \left(t\right)\geq w\right\},$$
we obtain that
$$\int_{\left[a,b\right]}n_{R,w}\circ \varphi d\mu =\int_{A_{\varphi }^{c}}\left(w-\varphi \right)d\mu =\int_{\left[a,b\right]}\left(w-\varphi \right)d\mu -\int_{A_{\varphi }}\left(w-\varphi \right)d\mu$$
$$=\int_{\left[a,b\right]}\left(w-\varphi \right)d\mu +\int_{\left[a,b\right]}p_{R,w}\circ \varphi d\mu .$$Thus, the conditions $\mu \left(\left[a,b\right]\right)=\nu \left(\left[a,b\right]\right)$, $\int_{\left[a,b\right]}\varphi d\mu =\int_{\left[a,b\right]}\psi d\nu$ and (Equation 7) imply that
$$\int_{\left[a,b\right]}n_{R,w}\circ \varphi d\mu =\int_{\left[a,b\right]}\left(w-\psi \right)d\nu +\int_{\left[a,b\right]}p_{R,w}\circ \varphi d\mu$$
$$\leq \int_{\left[a,b\right]}\left(w-\psi \right)d\nu +\int_{\left[a,b\right]}p_{R,w}\circ \psi d\nu$$
$$=\int_{\left[a,b\right]}\left(w-\psi \right)d\nu -\int_{A_{\psi }}\left(w-\psi \right)d\nu =\int_{\left[a,b\right]}n_{R,w}\circ \psi d\nu .$$The proof is complete. □

The next result contains integral majorization-type inequalities for some special functions.

**Lemma** **3.**
*Let $\left[a,b\right]\subset R$ with a<b, and let $\left(\left[a,b\right],A\right)$ be a measurable space, such that $B_{\left[a,b\right]}\subset A$. Suppose that one of the following two conditions is met:*

*(i) Let μ be a finite measure on A, let $\varphi :\left[a,b\right]\to R$ be a decreasing function, and let $\psi \in L\left(\left[a,b\right],\mu \right)$, such that (Equation 4) holds.*

*(ii) Let μ be a finite signed measure on A, and let φ, $\psi :\left[a,b\right]\to R$ be decreasing functions, such that (Equation 4) holds.*

*(a) If function f is either $id_{R}$ or $p_{R,w}$ for some w∈R, then*

(9)
$$\int_{\left[a,b\right]}f\circ \varphi d\mu \leq \int_{\left[a,b\right]}f\circ \psi d\mu .$$


*(b) Assume that*

(10)
$$\int_{\left[a,b\right]}\varphi d\mu =\int_{\left[a,b\right]}\psi d\mu$$

*is also satisfied. If $f=n_{R,w}$ for some w∈R, then inequality (Equation 9) holds too.*


**Proof.** We first consider the case where condition (i) is satisfied.(a) If $f=id_{R}$, then the result follows from (Equation 4).Now, assume that $f=p_{R,w}$ for some w∈R. Using the sets $A_{\varphi }$ and $A_{\psi }$ introduced in (Equation 8), we have
(11)$$\int_{\left[a,b\right]}f\circ \varphi d\mu =\int_{A_{\varphi }}\left(\varphi -w\right)d\mu \;\mathrm{and}\;\int_{\left[a,b\right]}f\circ \psi d\mu =\int_{A_{\psi }}\left(\psi -w\right)d\mu .$$Since φ is decreasing, either $A_{\varphi }=\left[a,c\right]$ or $A_{\varphi }=\left[a,c\right]$ for some $c\in \left[a,b\right]$. If $A_{\varphi }=Ø$, inequality (Equation 9) trivially follows from (Equation 11) and, thus, it can be supposed that $A_{\varphi }$ is a nonempty interval.Let $A_{\psi }^{c}$ denote the complement of $A_{\psi }$ with respect to $\left[a,b\right]$. Then, by the first part of (Equation 11) and Lemma 1 (a),
$$\int_{\left[a,b\right]}f\circ \varphi d\mu \leq \int_{A_{\varphi }}\left(\psi -w\right)d\mu =\int_{A_{\varphi }\cap A_{\psi }}\left(\psi -w\right)d\mu +\int_{A_{\varphi }\cap A_{\psi }^{c}}\left(\psi -w\right)d\mu ,$$
and, therefore, it follows from the definition of the set $A_{\psi }$ and from the second part of (Equation 11) that
$$\int_{\left[a,b\right]}f\circ \varphi d\mu \leq \int_{A_{\varphi }\cap A_{\psi }}\left(\psi -w\right)d\mu \leq \int_{A_{\psi }}\left(\psi -w\right)d\mu =\int_{\left[a,b\right]}f\circ \psi d\mu .$$(b) It comes from (a) and Lemma 2.We now turn to the case where condition (ii) is satisfied.(a) If $f=id_{R}$, then the result follows from (Equation 4).Now assume that $f=p_{R,w}$ for some w∈R.Using sets $A_{\varphi }$ and $A_{\psi }$ introduced in (Equation 8), we obtain that
(12)$$f\left(\psi \left(t\right)\right)-f\left(\varphi \left(t\right)\right)=\left\{\begin{array}{c} \psi \left(t\right)-\varphi \left(t\right),\;t\in A_{\varphi }⋂A_{\psi }\\ w-\varphi \left(t\right),\;t\in A_{\varphi }⋂A_{\psi }^{c}\\ \psi \left(t\right)-w,\;t\in A_{\varphi }^{c}⋂A_{\psi }\\ 0,\;t\in A_{\varphi }^{c}⋂A_{\psi }^{c} \end{array}\right.,$$
where any of the four intersections can be the empty set, their union is $\left[a,b\right]$, and at least one of the sets $A_{\varphi }⋂A_{\psi }^{c}$ and $A_{\varphi }^{c}⋂A_{\psi }$ is empty.We consider only the case when $A_{\varphi }⋂A_{\psi }^{c}=Ø$; that is, $A_{\varphi }\subset A_{\psi }$ (the other cases can be treated in a similar way). It can be supposed that the other three intersections are not empty. Since φ and ψ are decreasing, $A_{\varphi }$ and $A_{\psi }$ are nonempty intervals. It can be seen that $I_{1}:=A_{\varphi }$, $I_{2}:=A_{\varphi }^{c}⋂A_{\psi }$, and $I_{3}:=A_{\varphi }^{c}⋂A_{\psi }^{c}$ are pairwise disjoint and nonempty intervals with $I_{1}⋃I_{2}⋃I_{3}=\left[a,b\right]$. We define the function $\alpha :\left[a,b\right]\to R$ by
$$\alpha \left(t\right):=\left\{\begin{array}{c} 1,\;t\in I_{1}\\ \frac{\psi \left(t\right)-w}{\psi \left(t\right)-\varphi \left(t\right)},\;t\in I_{2}\\ 0,\;t\in I_{3} \end{array}\right.$$Then, α is well-defined and nonnegative. It is easy to verify that $\alpha \left(t\right)<1$ if $t\in I_{2}$.Next, we show that α is decreasing on $I_{2}$; that is, for all *t*, $s\in I_{2}$, t>s
$$\frac{\psi \left(t\right)-w}{\psi \left(t\right)-\varphi \left(t\right)}\geq \frac{\psi \left(s\right)-w}{\psi \left(s\right)-\varphi \left(s\right)}.$$This inequality is equivalent to
$$\left(\psi \left(t\right)-w\right)\left(w-\varphi \left(s\right)\right)\geq \left(\psi \left(s\right)-w\right)\left(w-\varphi \left(t\right)\right),$$
which is obvious.To summarize, we can see that α is decreasing.By (Equation 12) and the definition of α, we have
$$\int_{\left[a,b\right]}\left(f\circ \psi -f\circ \varphi \right)d\mu =\int_{\left[a,b\right]}\alpha \left(\psi -\varphi \right)d\mu ,$$
and, hence, Lemma 1 (b) can be applied.(b) It can be treated similarly to (b) under the condition of (i).The proof is complete. □

The next result is a simple consequence of the previous lemma.

**Corollary** **1.**
*Let $\left[a,b\right]\subset R$ with a<b, and let $\left(\left[a,b\right],A\right)$ be a measurable space, such that $B_{\left[a,b\right]}\subset A$. Suppose that one of the following two conditions is met:*

*(i) Let μ be a finite measure on A, let $\varphi \in L\left(\left[a,b\right],\mu \right)$, and let $\psi :\left[a,b\right]\to R$ be an increasing function, such that (Equation 4) holds.*

*(ii) Let μ be a finite signed measure on A, and let φ, $\psi :\left[a,b\right]\to R$ be increasing functions, such that (Equation 4) holds.*

*(a) If function f is either $-id_{R}$ or $n_{R,w}$ for some w∈R, then*

(13)
$$\int_{\left[a,b\right]}f\circ \varphi d\mu \geq \int_{\left[a,b\right]}f\circ \psi d\mu .$$


*(b) Assume that (Equation 10) is also satisfied. If $f=p_{R,w}$ for some w∈R, then inequality (Equation 13) holds too.*


**Proof.** Assume (i) is satisfied.(a) Under the conditions where −ψ is decreasing, $-\varphi \in L\left(\left[a,b\right],\mu \right)$, and
$$\int_{\left[a,x\right]}-\psi d\mu \leq \int_{\left[a,x\right]}-\varphi d\mu ,\;x\in \left[a,b\right].$$It now follows from Lemma 3 (a) that
$$\int_{\left[a,b\right]}f\circ \left(-\psi \right)d\mu \leq \int_{\left[a,b\right]}f\circ \left(-\varphi \right)d\mu ,$$
where *f* is either $id_{R}$ or $p_{R,w}$ for some w∈R. This gives the result by using ${\left(-a\right)}^{+}=a^{-}$.(b) By (Equation 10),
$$\int_{\left[a,b\right]}-\psi d\mu =\int_{\left[a,b\right]}-\varphi d\mu .$$Since ${\left(-a\right)}^{-}=a^{+}$, Lemma 3 (b) can be applied.We can prove it in a similar manner if (ii) is satisfied.The proof is complete. □

In the next statement, we will investigate the approximation of convex functions defined on intervals by monotone sequences of simple convex functions.

**Definition** **3.**
*Let C⊂R be an interval with the nonempty interior. A function f:C→R is called piecewise linear if it is continuous and there exists finite points $x_{1}<x_{2}<\ldots <x_{k}$ in the interior of C, such that the restriction of f to each interval $C⋂\left[-\infty ,x_{1}\right]$, $\left[x_{1},x_{2}\right]$, …, $C⋂\left[x_{k},\infty \right]$ is an affine function.*


**Theorem** **4.**
*Let C⊂R be an interval with a nonempty interior, and let f:C→R be a continuous convex function.*

*(a) Function f is the pointwise limit of an increasing sequence of piecewise linear convex functions on C.*

*(b) If f is increasing, then f is the pointwise limit of an increasing sequence of piecewise linear, increasing, and convex functions on C.*

*(c) If f is decreasing, then f is the pointwise limit of an increasing sequence of piecewise linear, decreasing, and convex functions on C.*

*(d) In all three cases, the convergence is uniform on every compact subinterval of C.*


**Proof.** (i) Assume first that *C* is a bounded interval with endpoints u<v.Let $y=l_{12}^{-}\left(x\right)$ be the equation of the left-hand tangent line to the graph of *f* at $\frac{u+v}{2}$, and let $y=l_{22}^{+}\left(x\right)$ be the equation of the right-hand tangent line to the graph of *f* at $\frac{u+v}{2}$. Define function $f_{1}:C\to R$ by
$$f_{1}\left(x\right):=\max\left(l_{12}^{+}\left(x\right),l_{22}^{-}\left(x\right)\right).$$It is obvious that $f_{1}$ is a simple convex function, it is increasing if *f* is increasing, it is decreasing if *f* is decreasing, and $f_{1}\leq f$.Next, we divide interval *C* into $2^{n}$ subintervals of equal widths for some n>1. If $u=:x_{0}<x_{1}<\ldots <x_{2^{n}}:=v$ is the appropriate partition, then let $y=l_{in}^{-}\left(x\right)$ be the equation of the left-hand tangent line to the graph of *f* at $x_{i}$, and let $y=l_{in}^{+}\left(x\right)$ be the equation of the right-hand tangent line to the graph of *f* at $x_{i}$
$\left(i=1,\ldots ,2^{n}-1\right)$. We define the function $f_{n}:C\to R$ by
$$f_{n}\left(x\right):=\max_{1\leq i\leq 2^{n}-1}\left(l_{in}^{-}\left(x\right),l_{in}^{+}\left(x\right)\right).$$It is also easy to believe that $f_{n}$ is a simple convex function; it is increasing if *f* is increasing, and it is decreasing if *f* is decreasing, $f_{n-1}\leq f_{n}\leq f$, and
$$f\left(x\right)-f_{n}\left(x\right)\leq \left(f_{-}^{\prime }\left(x_{2^{n}-1}\right)-f_{+}^{\prime }\left(x_{1}\right)\right)\frac{v-u}{2^{n}},\;x\in \left[x_{1},x_{2^{n}-1}\right].$$It can be seen that $\left(f_{n}\right)$ converges uniformly to *f* on every compact subinterval of the interior of *C* and, therefore, $\left(f_{n}\right)$ converges pointwise to *f* on the interior of *C*.Suppose that at least one of the endpoints belongs to *C*. We consider the case when v∈C. By the convexity of *f*,
$$\left(f\left(v\right)-f\left(x_{2^{n}-2}\right)\right)\frac{2^{n-1}}{v-u}\leq f_{+}^{\prime }\left(x_{2^{n}-1}\right)\leq \left(f\left(v\right)-f\left(x_{2^{n}-1}\right)\right)\frac{2^{n}}{v-u},$$
and, hence,
$$\frac{1}{2}\left(f\left(v\right)-f\left(x_{2^{n}-2}\right)\right)\leq f_{+}^{\prime }\left(x_{2^{n}-1}\right)\left(v-x_{2^{n}-1}\right)\leq f\left(v\right)-f\left(x_{2^{n}-1}\right),\;n\geq 2.$$Since *f* is continuous,
$$\lim_{n\to \infty }f_{+}^{\prime }\left(x_{2^{n}-1}\right)\left(v-x_{2^{n}-1}\right)=0,$$
and, thus, $f_{n}\left(v\right)\to f\left(v\right)$.(ii) Assume that *C* is an unbounded interval. We consider the case when *C* is bounded from the left with the left-hand endpoint u∈R. The other two cases can be treated in an analogous way.We can proceed similarly to the first part. Let n≥1 be an integer, and divide interval $C⋂\left[-\infty ,u+n\right]$ into $n2^{n}$ subintervals of equal width. If this partition is defined by the points $u=:x_{0}<x_{1}<\ldots <x_{n2^{n}}=u+n$, and equations $y=l_{in}^{-}\left(x\right)$ and $y=l_{in}^{+}\left(x\right)$ mean the same as in (i), then we define function $f_{n}:C\to R$ by
$$f_{n}\left(x\right):=\max_{1\leq i\leq n2^{n}-1}\left(l_{in}^{-}\left(x\right),l_{in}^{+}\left(x\right)\right).$$Then, $f_{n}$ is a simple convex function, it is increasing if *f* is increasing, it is decreasing if *f* is decreasing, and $f_{n-1}\leq f_{n}\leq f$.For all fixed k≥1, let ${\left(f_{k,n}\right)}_{n=1}^{\infty }$ be the sequence of functions constructed in (i) the restriction of *f* to $C⋂\left[-\infty ,u+k\right]$. It follows from the definitions of the introduced sequences of functions that for all n≥k≥1, the restriction of $f_{n}$ to $C⋂\left[-\infty ,u+k\right]$ is $f_{k,n}$. By part (i), this implies that $\left(f_{n}\right)$ converges pointwise to *f* on *C*.The proof is complete. □

**Remark** **1.**
*It is well known that if C⊂R is a compact interval with a nonempty interior, and f:C→R is a continuous convex function, then f is the pointwise limit of a sequence of piecewise linear convex functions on C. Its origins can be traced back to the paper by Popoviciu [11]. Our results can be applied to every continuous convex function defined on any type of interval, and the approximating sequence is increasing.*


**Remark** **2.**
*Let C⊂R be an interval with a nonempty interior, and let f:C→R be a piecewise linear convex function. If C is compact, then it is well known (see [1]) that f has a simple structure. The same is true for the functions described in Definition 3, and the proof can be copied as well. For the sake of completeness, and because we need representations in the proofs, we provide them.*

*(a) Function f has the following form:*

$$f\left(x\right)=\alpha x+\beta +\sum_{i=1}^{k}\gamma_{i}\left({\left(x-x_{i}\right)}^{+}+{\left(x-x_{i}\right)}^{-}\right),\;x\in C$$

*for suitable points $x_{1}<x_{2}<\ldots <x_{k}$ in the interior of C, α, β∈R, and $\gamma_{i}>0$
$\left(i=1,\ldots ,k\right)$.*

*(b) If f is increasing, then f is of the form*

$$f\left(x\right)=\alpha x+\beta +\sum_{i=1}^{k}\gamma_{i}{\left(x-x_{i}\right)}^{+},\;x\in C$$

*for suitable points $x_{1}<x_{2}<\ldots <x_{k}$ in the interior of C, α≥0, β∈R and $\gamma_{i}>0$
$\left(i=1,\ldots ,k\right)$.*

*(c) If f is decreasing, then f is of the form*

$$f\left(x\right)=\alpha x+\beta +\sum_{i=1}^{n}\gamma_{i}{\left(x-x_{i}\right)}^{-},\;x\in C$$

*for suitable points $x_{1}<x_{2}<\ldots <x_{k}$ in the interior of C, α≤0, β∈R and $\gamma_{i}>0$
$\left(i=1,\ldots ,k\right)$.*


The final result will be used to obtain Fejér-, especially Hermite-Hadamard type inequalities.

**Lemma** **4.**
*Let $\left[a,b\right]\subset R$ with a<b, and let μ be a finite signed measure on $B_{\left[a,b\right]}$ such that*

(14)
$$\mu \left(A\right)=\mu \left(a+b-A\right),\;A\in B_{\left[a,b\right]}.$$

*Assume φ, $\psi :\left[a,b\right]\to \left[a,b\right]$ are μ-integrable functions, such that*

(15)
$$\varphi \left(a+b-t\right)=a+b-\varphi \left(t\right),\;\psi \left(a+b-t\right)=a+b-\psi \left(t\right),\;t\in \left[a,b\right].$$


*(a) If*

(16)
$$\int_{\left[a,x\right]}\varphi d\mu \leq \int_{\left[a,x\right]}\psi d\mu ,\;x\in \left[a,\frac{a+b}{2}\right],$$

*then*

(17)
$$\int_{\left[a,x\right]}\varphi d\mu \leq \int_{\left[a,x\right]}\psi d\mu ,\;x\in \left[a,b\right]$$

*and*

(18)
$$\int_{\left[a,b\right]}\varphi d\mu =\int_{\left[a,b\right]}\psi d\mu =\frac{a+b}{2}\mu \left(\left[a,b\right]\right).$$


*(b) If μ is a measure and*

(19)
$$\varphi \left(t\right)\leq \psi \left(t\right),\;t\in \left[a,\frac{a+b}{2}\right],$$

*then (Equation 16) holds.*


**Proof.** (a) We divide the proof into six parts.(i) We define function $T:\left[a,\frac{a+b}{2}\right]\to \left[\frac{a+b}{2},b\right]$ by $T\left(t\right):=a+b-t$. Let $T\left(\mu \right)$ be the image measure of the restriction of μ to $B_{\left[a,\frac{a+b}{2}\right]}$ under the mapping *T*. If $B\in B_{\left[\frac{a+b}{2},b\right]}$, then by (Equation 14),
$$\mu \left(T^{-1}\left(B\right)\right)=\mu \left(a+b-B\right)=\mu \left(B\right),$$
and, hence, $T\left(\mu \right)$ is the restriction of μ to $B_{\left[\frac{a+b}{2},b\right]}$.(ii) Since
$$\mu \left(\left[a,\frac{a+b}{2}\right]\right)=\mu \left(\left[\frac{a+b}{2},b\right]\right),$$
it follows that
(20)$$\frac{a+b}{2}\mu \left(\left\{\frac{a+b}{2}\right\}\right)+\left(a+b\right)\mu \left(\left[a,\frac{a+b}{2}\right]\right)=\frac{a+b}{2}\mu \left(\left[a,b\right]\right).$$According to (Equation 15),
$$\varphi \left(\frac{a+b}{2}\right)=\psi \left(\frac{a+b}{2}\right)=\frac{a+b}{2}.$$For the rest of the proof of (a), assume $x\in \left[\frac{a+b}{2},b\right]$.(iii) By (i) and the first part of (Equation 15),
$$\int_{\left[a,x\right]}\varphi d\mu =\int_{\left[a,\frac{a+b}{2}\right]}\varphi d\mu +\int_{\left[\frac{a+b}{2},x\right]}\varphi d\mu =\int_{\left[a,\frac{a+b}{2}\right]}\varphi d\mu +\int_{\left[\frac{a+b}{2},x\right]}\varphi dT\left(\mu \right)$$
$$=\int_{\left[a,\frac{a+b}{2}\right]}\varphi d\mu +\int_{\left[a+b-x,\frac{a+b}{2}\right]}\varphi \circ Td\mu =\int_{\left[a,\frac{a+b}{2}\right]}\varphi d\mu +\int_{\left[a+b-x,\frac{a+b}{2}\right]}\left(a+b-\varphi \right)d\mu$$
(21)$$=\int_{\left[a,a+b-x\right]}\varphi d\mu +\frac{a+b}{2}\mu \left(\left\{\frac{a+b}{2}\right\}\right)+\left(a+b\right)\mu \left(\left[a+b-x,\frac{a+b}{2}\right]\right).$$By using the second part of (Equation 15), we can similarly obtain that
(22)$$\int_{\left[a,x\right]}\psi d\mu =\int_{\left[a,a+b-x\right]}\psi d\mu +\frac{a+b}{2}\mu \left(\left\{\frac{a+b}{2}\right\}\right)+\left(a+b\right)\mu \left(\left[a+b-x,\frac{a+b}{2}\right]\right).$$(iv) Since $a+b-x\in \left[a,\frac{a+b}{2}\right]$, (Equation 16) and Lemma 1 (a) yield that
$$\int_{\left[a,a+b-x\right]}\varphi d\mu \leq \int_{\left[a,a+b-x\right]}\psi d\mu .$$(v) It can be seen that (iv), (Equation 21) and (Equation 22) imply inequality (Equation 17).(vi) By applying (Equation 21) and (Equation 22) to x=b, (Equation 18) follows from (Equation 20).(b) According to the nonnegativity of μ and (Equation 19), inequality (Equation 16) obviously holds.The proof is complete. □

## 3. Majorization-Type Theorems for Integrals

The key to a further discussion lies in the following result.

By $N_{+}$ we denote the set of positive integers.

The interior of a set H⊂R is denoted by $H^{\circ }$.

**Theorem** **5.**
*Let $\left[a,b\right]\subset R$ with a<b, and let $\left(\left[a,b\right],A\right)$ be a measurable space, such that $B_{\left[a,b\right]}\subset A$ and μ, ν are finite signed measures on A. Let C⊂R be an interval with a nonempty interior, and let φ, $\psi :\left[a,b\right]\to C$ be functions, such that $\varphi \in L\left(\left[a,b\right],\mu \right)$ and $\psi \in L\left(\left[a,b\right],\nu \right)$.*

*(a) Let $F_{C}^{i}$ denote the set of all increasing and convex functions f:C→R for which $f\circ \varphi \in L\left(\left[a,b\right],\mu \right)$ and $f\circ \psi \in L\left(\left[a,b\right],\nu \right)$. Then, for each $f\in F_{C}^{i}$ inequality,*

(23)
$$\int_{\left[a,b\right]}f\circ \varphi d\mu \leq \int_{\left[a,b\right]}f\circ \psi d\nu$$

*holds if and only if $\mu \left(\left[a,b\right]\right)=\nu \left(\left[a,b\right]\right)$ and it is satisfied in the following special cases: function f is either $id_{C}$ or $p_{C,x}$
$\left(x\in C^{\circ }\right)$.*

*(b) Let $F_{C}^{d}$ denote the set of all decreasing and convex functions f:C→R for which $f\circ \varphi \in L\left(\left[a,b\right],\mu \right)$ and $f\circ \psi \in L\left(\left[a,b\right],\nu \right)$. Then, for each $f\in F_{C}^{d}$ inequality, (Equation 23) holds if and only if $\mu \left(\left[a,b\right]\right)=\nu \left(\left[a,b\right]\right)$ and it is satisfied in the following special cases: the function f is either $-id_{C}$ or $n_{C,x}$
$\left(x\in C^{\circ }\right)$.*

*(c) Let $F_{C}$ denote the set of all convex functions f:C→R for which $f\circ \varphi \in L\left(\left[a,b\right],\mu \right)$ and $f\circ \psi \in L\left(\left[a,b\right],\nu \right)$. Then, for each $f\in F_{C}$ inequality, (Equation 23) holds if and only if $\mu \left(\left[a,b\right]\right)=\nu \left(\left[a,b\right]\right)$ and it is satisfied in the following special cases: the function f is either $id_{C}$ or $-id_{C}$ or $p_{C,x}$
$\left(x\in C^{\circ }\right)$.*


**Proof.** We first note that if inequality (Equation 23) holds for each $f\in F_{C}^{i}$, then $\int_{\left[a,b\right]}\varphi d\mu \leq \int_{\left[a,b\right]}\psi d\nu$, if (Equation 23) holds for each $f\in F_{C}^{d}$, then $\int_{\left[a,b\right]}\varphi d\mu \geq \int_{\left[a,b\right]}\psi d\nu$, and if (Equation 23) holds for each $f\in F_{C}$, then
(24)$$\int_{\left[a,b\right]}\varphi d\mu =\int_{\left[a,b\right]}\psi d\nu .$$(a) The constant functions $f_{1}$, $f_{2}:C\to R$, $f_{1}\left(x\right):=1$ and $f_{2}\left(x\right):=-1$ belong to $F_{C}^{i}$ and, hence, (Equation 23) implies $\mu \left(\left[a,b\right]\right)=\nu \left(\left[a,b\right]\right)$. The functions $id_{C}$ and $p_{C,x}$
$\left(x\in C^{\circ }\right)$ are increasing and convex, and since $\varphi \in L\left(\left[a,b\right],\mu \right)$, $\psi \in L\left(\left[a,b\right],\nu \right)$ and μ, ν are finite, they belong to $F_{C}^{i}$. This shows that the condition is necessary.To prove sufficiency, we distinguish two cases.(i) Assume that *f* is continuous.By Theorem 4 (b), *f* is the pointwise limit of an increasing sequence of piecewise linear, increasing, and convex functions on *C*. If $\left(f_{n}\right)$ is such a sequence, then $\left(f_{n}\circ \varphi \right)$ converges pointwise to f∘φ and $\left(f_{n}\circ \psi \right)$ converges pointwise to f∘ψ.By Remark 2 (b), if *g* is a piecewise linear increasing and convex function on *C*, then *g* is of the form
(25)$$g\left(x\right)=\alpha x+\beta +\sum_{i=1}^{k}\gamma_{i}{\left(x-x_{i}\right)}^{+},\;x\in C$$
for suitable points $x_{1}<x_{2}<\ldots <x_{k}$ in the interior of *C*, α≥0, β∈R and $\gamma_{i}>0$
$\left(i=1,\ldots ,k\right)$. Since $\varphi \in L\left(\left[a,b\right],\mu \right)$ and μ is finite, $g\circ \varphi \in L\left(\left[a,b\right],\mu \right)$. Similarly, $g\circ \psi \in L\left(\left[a,b\right],\nu \right)$ and, hence, $g\in F_{C}^{i}$.Since
(26)$$\left|f_{n}\circ \varphi \right|\leq \max\left(\left|f\circ \varphi \right|,\left|f_{1}\circ \varphi \right|\right),\;\left|f_{n}\circ \psi \right|\leq \max\left(\left|f\circ \psi \right|,\left|f_{1}\circ \psi \right|\right),$$
the dominated convergence theorem implies that
$$\int_{\left[a,b\right]}f_{n}\circ \varphi d\mu \to \int_{\left[a,b\right]}f\circ \varphi d\mu \;\mathrm{and}\;\int_{\left[a,b\right]}f_{n}\circ \psi d\nu \to \int_{\left[a,b\right]}f\circ \psi d\nu .$$In summary, it is enough to prove (Equation 23) for piecewise linear increasing and convex functions on *C*. Since such a function is of the form (Equation 25), it follows from the condition.(ii) Assume that *f* is not continuous at the right-hand endpoint of the interval *C*.Then, it is not hard to believe that there exists a decreasing sequence ${\left(f_{n}\right)}_{n=1}^{\infty }$ from $F_{C}^{i}$, such that $f_{n}$ is continuous $\left(n\in N_{+}\right)$ and $\left(f_{n}\right)$ converges pointwise to *f* on *C*. In this case, (Equation 26) is also satisfied and, therefore, the result follows from the first part of the proof and the dominated convergence theorem.(b) It can be proven similarly to (a) by using Theorem 4 (c) and taking Remark 2 (c) into account.(c) It can be proven similarly to (a) by using Theorem 4 (a) and taking Remark 2 (a) into account. In the sufficiency part of the proof, we can apply Lemma 2, which shows that (Equation 23) holds for $n_{C,x}$ $\left(x\in C^{\circ }\right)$ too.The proof is complete. □

**Remark** **3.**
*(a) By Lemma 2, in part (c) of Theorem 5, “the function f is either $id_{C}$ or $-id_{C}$ or $p_{C,x}$
$\left(x\in C^{\circ }\right)$” can be replaced by “function f is either $id_{C}$ or $-id_{C}$ or $n_{C,x}$
$\left(x\in C^{\circ }\right)$”.*

*(b) It is easy to verify that Theorem 5 (c) contains the following result from Levin and Stečkin [12]: Let $H:\left[a,b\right]\to R$ be a function with bounded variations, such that $H\left(a\right)=0$. Then*

$$\int_{\left[a,b\right]}fdH\geq 0$$

*for all continuous and convex functions on $\left[a,b\right]$ if and only if the following three conditions are fulfilled:*

$$H\left(b\right)=0,\;\int_{\left[a,b\right]}H\left(x\right)dx=0,\;\int_{\left[a,x\right]}H\left(t\right)dt\geq 0,\;x\in \left[a,b\right].$$


*(c) It can be easily seen that the main results of Theorem 6 and Theorem 7 in the paper by Barnett, Cerone, and Dragomir [8] are also special cases of Theorem 5 (c). They provide some sufficient conditions for the inequality*

$$\int_{a}^{b}p\left(t\right)f\left(\varphi \left(t\right)\right)du\left(t\right)\leq \int_{a}^{b}p\left(t\right)f\left(\psi \left(t\right)\right)du\left(t\right),$$

*to be valid, where f is a convex function, p is a bounded variation on $\left[a,b\right]$ and is nonnegative, u is increasing, and the Stietjes integral is used. Their proofs are specific; The notions of sub-differential and a Chebyshev-type inequality are used.*


The next result is a special case of Theorem 5. It more closely follows the usual form of majorization inequalities for integrals.

**Theorem** **6.**
*Let C⊂R be an interval with a nonempty interior, and let f:C→R be a convex function. Let $\left[a,b\right]\subset R$ with a<b, and let $\left(\left[a,b\right],A\right)$ be a measurable space, such that $B_{\left[a,b\right]}\subset A$.*

*(a) Suppose that one of the following two conditions is met:*

*(i) Let μ be a finite measure on A. Assume $\varphi :\left[a,b\right]\to C$ is a decreasing function, and $\psi :\left[a,b\right]\to C$ is a μ-integrable function for which f∘ψ is also μ-integrable.*

*(ii) Let μ be a finite signed measure on A. Assume φ, $\psi :\left[a,b\right]\to C$ are decreasing functions.*

*(a${}_{1}$) If f is increasing and (Equation 4) is satisfied, then*

(27)
$$\int_{\left[a,b\right]}f\circ \varphi d\mu \leq \int_{\left[a,b\right]}f\circ \psi d\mu .$$


*(a${}_{2}$) If (Equation 4) and (Equation 10) are satisfied, then inequality (Equation 27) holds too.*

*(b) Suppose that one of the following two conditions is met:*

*(i) Let μ be a finite measure on A. Assume $\varphi :\left[a,b\right]\to C$ is a μ-integrable function for which f∘φ is also μ-integrable, and $\psi :\left[a,b\right]\to C$ is an increasing function.*

*(ii) Let μ be a finite signed measure on A. Assume φ, $\psi :\left[a,b\right]\to C$ are increasing functions.*

*(b${}_{1}$) If f is decreasing and (Equation 4) is satisfied, then*

(28)
$$\int_{\left[a,b\right]}f\circ \varphi d\mu \geq \int_{\left[a,b\right]}f\circ \psi d\mu .$$


*(b${}_{2}$) If (Equation 4) and (Equation 10) are satisfied, then inequality (Equation 28) holds too.*


**Proof.** (a) The proof is valid even under conditions (i) and (ii).The functions φ, ψ, f∘φ, and f∘ψ are obviously μ-integrable.(a${}_{1}$) It follows from Theorem 5 (a) by applying Lemma 3 (a).(a${}_{2}$) It can be proven similarly to (a${}_{1}$) by using Theorem 5 (c) and taking into account Lemma 3.(b${}_{1}$) It can be proven similarly to (a${}_{1}$) by using Theorem 5 (b) and taking into account Corollary 1 (a).(b${}_{2}$) It can be proven similarly to (a${}_{1}$) by using Theorem 5 (c) and taking into account Corollary 1.The proof is complete. □

It is worth mentioning the following two special cases of Theorem 6 separately.

First, we consider the case when μ is absolutely continuous with respect to a σ-finite measure ν on A. In this case, μ has a ν-almost-everywhere uniquely determined density $p:\left[a,b\right]\to R$, with respect to ν. Since μ is finite, *p* is ν-integrable.

**Corollary** **2.**
*Let C⊂R be an interval with a nonempty interior, and let f:C→R be a convex function. Let $\left[a,b\right]\subset R$ with a<b, let $\left(\left[a,b\right],A,\nu \right)$ be a measure space, such that $B_{\left[a,b\right]}\subset A$, and ν is a σ-finite measure ν on A, and let $p:\left[a,b\right]\to R$ be a ν-integrable function.*

*(a) Suppose that one of the following two conditions is met:*

*(i) Assume p is nonnegative, $\varphi :\left[a,b\right]\to C$ is a decreasing function, and $\psi :\left[a,b\right]\to C$ is an A-measurable function for which ψp and $\left(f\circ \psi \right)p$ are ν-integrable.*

*(ii) Assume that φ, $\psi :\left[a,b\right]\to C$ are decreasing functions.*

*(a${}_{1}$) If f is increasing and*

(29)
$$\int_{\left[a,x\right]}\varphi pd\nu \leq \int_{\left[a,x\right]}\psi pd\nu ,\;x\in \left[a,b\right]$$

*is satisfied, then*

(30)
$$\int_{\left[a,b\right]}\left(f\circ \varphi \right)pd\nu \leq \int_{\left[a,b\right]}\left(f\circ \psi \right)pd\nu .$$


*(a${}_{2}$) If (Equation 29) and*

(31)
$$\int_{\left[a,b\right]}\varphi pd\nu =\int_{\left[a,b\right]}\psi pd\nu$$

*are satisfied, then inequality (Equation 30) holds too.*

*(b) Suppose that one of the following two conditions is met:*

*(i) Assume p is nonnegative, $\varphi :\left[a,b\right]\to C$ is a A-measurable function for which φp and $\left(f\circ \varphi \right)p$ re ν-integrable, and $\psi :\left[a,b\right]\to C$ is an increasing function.*

*(ii) Assume φ, $\psi :\left[a,b\right]\to C$ are increasing functions.*

*(b${}_{1}$) If f is decreasing and (Equation 29) is satisfied, then*

(32)
$$\int_{\left[a,b\right]}\left(f\circ \varphi \right)pd\nu \geq \int_{\left[a,b\right]}\left(f\circ \psi \right)pd\nu .$$


*(b${}_{2}$) If (Equation 29) and (Equation 31) are satisfied, then inequality (Equation 32) holds too.*


**Proof.** Let the set function μ be defined on A by
$$\mu \left(A\right)=\int_{A}pd\nu ,\;A\in A.$$If *p* is nonnegative, then μ is a measure on A; otherwise, μ is a signed measure on A.The result immediately follows from Theorem 6.The proof is complete. □

This is an important special case of the previous result when $A=L_{\left[a,b\right]}$, and μ is absolutely continuous with respect to the Lebesgue measure λ on $L_{\left[a,b\right]}$.

Next, we consider the case when μ is a discrete measure on A.

**Corollary** **3.**
*Let C⊂R be an interval with a nonempty interior, and let f:C→R be a convex function. Let the index set I be either a finite set of the form $\left\{1,\ldots ,n\right\}$ for some integer n≥1 or $N_{+}$. Let ${\left(\mu_{i}\right)}_{i\in I}$ be a sequence of real numbers with $\sum_{i\in I}\left|\mu_{i}\right|<\infty$.*

*(a) Suppose that one of the following two conditions is met:*

*(i) Assume $\mu_{i}\geq 0$
$\left(i\in I\right)$, ${\left(x_{i}\right)}_{i\in I}$ is a decreasing sequence in C, and ${\left(y_{i}\right)}_{i\in I}$ is a sequence in C for which the series $\sum_{i\in I}y_{i}\mu_{i}$
$\sum_{i\in I}f\left(y_{i}\right)\mu_{i}$ are absolutely convergent.*

*(ii) Assume ${\left(x_{i}\right)}_{i\in I}$ and ${\left(y_{i}\right)}_{i\in I}$ are decreasing sequences in C.*

*(a${}_{1}$) If f is increasing and*

(33)
$$\sum_{i=1}^{k}x_{i}\mu_{i}\leq \sum_{i=1}^{k}y_{i}\mu_{i},\;k\in I$$

*is satisfied, then*

(34)
$$\sum_{i\in I}f\left(x_{i}\right)\mu_{i}\leq \sum_{i\in I}f\left(y_{i}\right)\mu_{i}.$$


*(a${}_{2}$) If (Equation 33) and*

(35)
$$\sum_{i\in I}x_{i}\mu_{i}=\sum_{i\in I}y_{i}\mu_{i},$$

*are satisfied, then inequality (Equation 34) holds too.*

*(b) Suppose that one of the following two conditions is met:*

*(i) Assume $\mu_{i}\geq 0$
$\left(i\in I\right)$, ${\left(x_{i}\right)}_{i\in I}$ is a sequence in C for which the series $\sum_{i\in I}x_{i}\mu_{i}$ and $\sum_{i\in I}f\left(x_{i}\right)\mu_{i}$ are absolutely convergent, and ${\left(y_{i}\right)}_{i\in I}$ is an increasing sequence in C.*

*(ii) Assume ${\left(x_{i}\right)}_{i\in I}$ and ${\left(y_{i}\right)}_{i\in I}$ are increasing sequences in C.*

*(b${}_{1}$) If f is decreasing and (Equation 33) is satisfied, then*

(36)
$$\sum_{i\in I}f\left(x_{i}\right)\mu_{i}\geq \sum_{i\in I}f\left(y_{i}\right)\mu_{i}.$$


*(b${}_{2}$) If (Equation 33) and (Equation 35) are satisfied, then inequality (Equation 36) holds too.*


**Proof.** Let $\left[a,b\right]\subset R$ with a<b, let ${\left(t_{i}\right)}_{i\in I}$ be a strictly decreasing sequence in $\left[a,b\right]$, and let the measure μ be defined on $B_{\left[a,b\right]}$ by
$$\mu :=\sum_{i\in I}\mu_{i}\varepsilon_{t_{i}},$$
where the measure $\varepsilon_{t_{i}}$ on $B_{\left[a,b\right]}$ is the unit mass at $t_{i}$
$\left(i\in I\right)$. Since $\sum_{i\in I}\left|\mu_{i}\right|<\infty$, μ is a finite set function.If $\mu_{i}\geq 0$
$\left(i\in I\right)$, then μ is a measure on $B_{\left[a,b\right]}$; otherwise, μ is a signed measure on $B_{\left[a,b\right]}$.(a) Under condition (i), it is not hard to verify that there exist functions $\varphi ,\psi :\left[a,b\right]\to C$, such that φ is continuous and decreasing, ψ is Borel-measurable, and
$$\varphi \left(t_{i}\right)=x_{i},\;\psi \left(t_{i}\right)=y_{i},\;i\in I.$$If (ii) holds, then ψ can also be chosen as a continuous and decreasing function.We can apply Theorem 6 (a).(b) It can be verified in a similar manner as (a).The proof is complete. □

**Remark** **4.**
*The result contains the weighted version of the Hardy–Littlewood–Pólya inequality and Fuchs inequality (see Theorem 1 and Theorem 2), and even extends them to countably infinite sequences.*


## 4. Hermite-Hadamard-Fejér-Type Inequalities

The first statement includes known results in a single framework.

**Theorem** **7.**
*Let $\left[a,b\right]\subset R$ with a<b, and let μ be a finite signed measure on $B_{\left[a,b\right]}$, such that $\mu \left(\left[a,b\right]\right)>0$.*

*(a) The inequality*

(37)
$$f\left(x_{\mu }\right)\mu \left(\left[a,b\right]\right)\leq \int_{\left[a,b\right]}fd\mu$$

*holds for some $x_{\mu }\in \left[a,b\right]$ and all convex functions $f:\left[a,b\right]\to R$ if and only if*

(38)
$$x_{\mu }:=\frac{1}{\mu \left(\left[a,b\right]\right)}\int_{\left[a,b\right]}td\mu \left(t\right)$$

*and*

(39)
$$\int_{\left[a,x\right]}\left(x-t\right)d\mu \left(t\right)\geq 0\;\mathrm{and}\;\int_{\left[x,b\right]}\left(t-x\right)d\mu \left(t\right)\geq 0,\;x\in \left[a,b\right].$$


*(b) Assume $x_{\mu }\in \left[a,b\right]$. The inequality*

(40)
$$\int_{\left[a,b\right]}fd\mu \leq \left(\frac{b-x_{\mu }}{b-a}f\left(a\right)+\frac{x_{\mu }-a}{b-a}f\left(b\right)\right)\mu \left(\left[a,b\right]\right)$$

*holds for all convex functions $f:\left[a,b\right]\to R$ if and only if*

(41)
$$\frac{b-x}{b-a}\int_{\left[a,x\right]}\left(t-a\right)d\mu \left(t\right)+\frac{x-a}{b-a}\int_{\left[x,b\right]}\left(b-t\right)d\mu \left(t\right)\geq 0,\;x\in \left[a,b\right].$$



**Proof.** (a) By Theorem 5 (c), the assertion is true if and only if
(42)$$x_{\mu }\cdot \mu \left(\left[a,b\right]\right)=\int_{\left[a,b\right]}td\mu \left(t\right),$$
(43)$$\int_{\left[a,b\right]}p_{\left[a,b\right],x}\left(x_{\mu }\right)d\mu \leq \int_{\left[a,b\right]}p_{\left[a,b\right],x}d\mu ,\;x\in \left[a,b\right]$$
and
(44)$$\int_{\left[a,b\right]}n_{\left[a,b\right],x}\left(x_{\mu }\right)d\mu \leq \int_{\left[a,b\right]}n_{\left[a,b\right],x}d\mu ,\;x\in \left[a,b\right]$$
are satisfied.By $\mu \left(\left[a,b\right]\right)>0$, (Equation 38) is equivalent to (Equation 42).It is obvious that $x_{\mu }\in \left[a,b\right]$ is equivalent to
$$\int_{\left[a,b\right]}\left(b-t\right)d\mu \left(t\right)\geq 0\;\mathrm{and}\;\int_{\left[a,b\right]}\left(t-a\right)d\mu \left(t\right)\geq 0.$$By elementary calculations, we can obtain that inequalities (Equation 43) and (Equation 44) hold exactly if
(45)$$\left\{\begin{array}{c} 0\leq \int_{\left[x,b\right]}\left(t-x\right)d\mu \left(t\right),\;\mathrm{if}\;x\in \left[x_{\mu },b\right]\\ 0\leq \int_{\left[a,x\right]}\left(x-t\right)d\mu \left(t\right),\;\mathrm{if}\;x\in \left[a,x_{\mu }\right] \end{array}\right..$$The remaining task is to prove that (Equation 45) implies (Equation 39). Since
$$\int_{\left[x,b\right]}\left(t-x\right)d\mu \left(t\right)=\mu \left(\left[a,b\right]\right)\left(x_{\mu }-x\right)+\int_{\left[a,x\right]}\left(x-t\right)d\mu \left(t\right),\;x\in \left[a,b\right],$$
the first inequality in (Equation 39) follows from $\mu \left(\left[a,b\right]\right)>0$ and (Equation 45). The second inequality in (Equation 39) can be handled in a similar way.(b) Let the function $\varphi_{l}:\left[a,b\right]\to \left[a,b\right]$ be defined by
$$\varphi_{l}\left(t\right)=\left\{\begin{array}{c} b,\;\mathrm{if}\;t\in \left[a,x_{\mu }\right]\\ x_{\mu },\;\mathrm{if}\;t=x_{\mu }\\ a,\;\mathrm{if}\;t\in \left[x_{\mu },b\right] \end{array}\right.,$$
and introduce the measure $\hat{\lambda }:=\frac{\mu \left(\left[a,b\right]\right)}{b-a}\lambda$ on $B_{\left[a,b\right]}$.Then
$$\int_{\left[a,b\right]}\varphi_{l}d\hat{\lambda }=\int_{\left[a,b\right]}td\mu \left(t\right),$$
and
$$\int_{\left[a,b\right]}f\circ \varphi_{l}d\hat{\lambda }=\left(\frac{b-x_{\mu }}{b-a}f\left(a\right)+\frac{x_{\mu }-a}{b-a}f\left(b\right)\right)\mu \left(\left[a,b\right]\right)$$
for all convex functions $f:\left[a,b\right]\to R$.It now follows from Theorem 5 (c) that inequality (Equation 40) holds for all convex functions $f:\left[a,b\right]\to R$ if and only if
$$\int_{\left[a,b\right]}p_{\left[a,b\right],x}d\mu \leq \int_{\left[a,b\right]}p_{\left[a,b\right],x}\circ \varphi_{l}d\hat{\lambda },\;x\in \left[a,b\right]$$
and
$$\int_{\left[a,b\right]}n_{\left[a,b\right],x}d\mu \leq \int_{\left[a,b\right]}n_{\left[a,b\right],x}\circ \varphi_{l}d\hat{\lambda },\;x\in \left[a,b\right]$$
are satisfied; however, some easy calculations show that both inequalities are equivalent to (Equation 41).The proof is complete. □

**Remark** **5.**
*(a) The number $x_{\mu }$ defined in (Equation 38) is called the barycenter of μ.*

*(b) The part (a) of Theorem 7 was discovered by Fink [9]. The idea of his proof is different from the one we use; it is based on the integral representation of convex functions. Finite signed measures on $B_{\left[a,b\right]}$, for which the measure of $\left[a,b\right]$ is positive and (Equation 39) holds, are called Steffensen–Popoviciu measures.*

*(c) In [9], Fink also presented a sufficient but not necessary condition for the satisfaction of inequality (Equation 40). Part (b) of Theorem 7, which is the complete characterization of the measures for which (Equation 40) holds, is given by Florea and Niculescu in [10]. Their proof is a modification of Fink’s argument, which is based on the integral representation of twice continuously differentiable convex functions using the Green function of the operator $\frac{d^{2}}{dx^{2}}$ with homogeneous boundary conditions $y\left(a\right)=y\left(b\right)=0$. This is also different from the method we follow.*

*(d) We emphasize that the same natural technique is used to prove Theorem 7 (a) and (b). This may be new.*

*(e) Condition (Equation 41) does not imply $x_{\mu }\in \left[a,b\right]$ in general. This can be illustrated by elementary examples.*

*(f) For the sake of completeness, we provide examples of measures that satisfy exactly one of the following conditions: (Equation 39) or (Equation 41).*

*(i) If the measure μ on $B_{\left[0,3\right]}$ is defined by*

$$\mu :=2\varepsilon_{0}-\varepsilon_{1}-\varepsilon_{2}+\varepsilon_{3},$$

*then some straightforward calculation shows that condition (Equation 39) is satisfied, but (Equation 41) does not hold.*

*In this case, the barycenter of μ is 0, and inequality (Equation 37) has the form*

$$f\left(0\right)\leq 2f\left(0\right)-f\left(1\right)-f\left(2\right)+f\left(3\right)$$

*which is obviously fulfilled by the convexity of f. The form of inequality (Equation 40) is*

$$2f\left(0\right)-f\left(1\right)-f\left(2\right)+f\left(3\right)\leq f\left(0\right)$$

*which is not true in general.*

*(ii) If the measure μ on $B_{\left[0,2\right]}$ is defined by*

$$\mu :=-\varepsilon +2\varepsilon_{1}+2\varepsilon_{2},$$

*then it is also easy to show that condition (Equation 41) is satisfied, but (Equation 39) does not hold.*

*Now, the barycenter of μ is 2, and inequality (Equation 37) has the form*

$$f\left(2\right)\leq -f\left(0\right)+2f\left(1\right)+2f\left(2\right)$$

*which does not hold in general. The form of inequality (Equation 40) is*

$$-f\left(0\right)+2f\left(1\right)+2f\left(2\right)\leq 3f\left(2\right)$$

*which comes from the convexity of f.*

*(g) It follows from Theorem 7 (a) and (b) that inequalities*

(46)
$$f\left(x_{\mu }\right)\leq \frac{1}{\mu \left(\left[a,b\right]\right)}\int_{\left[a,b\right]}fd\mu \leq \frac{b-x_{\mu }}{b-a}f\left(a\right)+\frac{x_{\mu }-a}{b-a}f\left(b\right).$$

*are satisfied for all convex functions $f:\left[a,b\right]\to R$ if and only if both conditions (Equation 39) and (Equation 41) are true. It is still an open question on how to write up the joint fulfillment of conditions (Equation 39) and (Equation 41) in a compact form.*


In the next result, we deal with refinements of inequalities given in (Equation 46).

**Theorem** **8.**
*Let $\left[a,b\right]\subset R$ with a<b, and let μ be a finite measure on $B_{\left[a,b\right]}$, such that $\mu \left(\left[a,b\right]\right)>0$. Assume $\varphi_{1}$, $\varphi_{0}$, $\psi_{1}$, $\psi_{0}:\left[a,b\right]\to \left[a,b\right]$ are increasing functions, such that*

(47)
$$\int_{\left[a,x\right]}\varphi_{0}d\mu \leq \int_{\left[a,x\right]}\varphi_{1}d\mu \leq \int_{\left[a,x\right]}td\mu \left(t\right)\leq \int_{\left[a,x\right]}\psi_{1}d\mu \leq \int_{\left[a,x\right]}\psi_{0}d\mu ,\;x\in \left[a,b\right]$$

*and*

(48)
$$\int_{\left[a,b\right]}\varphi_{0}d\mu =\int_{\left[a,b\right]}\varphi_{1}d\mu =\int_{\left[a,b\right]}\psi_{1}d\mu =\int_{\left[a,b\right]}\psi_{0}d\mu =\int_{\left[a,b\right]}td\mu \left(t\right)$$

*are satisfied. Then, for all convex functions $f:\left[a,b\right]\to R$, we have*

(49)
$$f\left(x_{\mu }\right)\mu \left(\left[a,b\right]\right)$$


(50)
$$\leq \int_{\left[a,b\right]}f\circ \psi_{0}d\mu \leq \int_{\left[a,b\right]}f\circ \psi_{1}d\mu \leq \int_{\left[a,b\right]}fd\mu \leq \int_{\left[a,b\right]}f\circ \varphi_{1}d\mu \leq \int_{\left[a,b\right]}f\circ \varphi_{0}d\mu$$


(51)
$$\leq \left(\frac{b-x_{\mu }}{b-a}f\left(a\right)+\frac{x_{\mu }-a}{b-a}f\left(b\right)\right)\mu \left(\left[a,b\right]\right).$$



**Proof.** Inequalities in (Equation 50) are immediate consequences of Theorem 6 (b${}_{2}$).To prove (Equation 49), introduce the increasing function $\psi_{u}:\left[a,b\right]\to \left[a,b\right]$, $\psi_{u}\left(t\right):=x_{\mu }$.By Theorem 6 (b${}_{2}$), it is enough to show that
(52)$$\int_{\left[a,x\right]}\psi_{0}d\mu \leq \int_{\left[a,x\right]}\psi_{u}d\mu ,\;x\in \left[a,b\right].$$We argue indirectly and suppose there exists an $x\in \left[a,b\right]$, such that (Equation 52) does not hold. Since $\psi_{0}$ is increasing, it follows that
(53)$$x_{\mu }\cdot \mu \left(\left[a,x\right]\right)<\int_{\left[a,x\right]}\psi_{0}d\mu \leq \psi_{0}\left(x\right)\mu \left(\left[a,x\right]\right).$$ The strict inequality in (Equation 53) implies that $\mu \left(\left[a,x\right]\right)>0$ and, hence, $x_{\mu }<\psi_{0}\left(x\right)$. Since $\psi_{0}$ is increasing, this and the firs part of (Equation 53) yield that
$$\int_{\left[a,b\right]}\psi_{0}d\mu =\int_{\left[a,x\right]}\psi_{0}d\mu +\int_{\left[x,b\right]}\psi_{0}d\mu >x_{\mu }\cdot \mu \left(\left[a,x\right]\right)+x_{\mu }\cdot \mu \left(\left[x,b\right]\right)$$
$$=x_{\mu }\cdot \mu \left(\left[a,b\right]\right)=\int_{\left[a,b\right]}td\mu \left(t\right)$$
which contradicts (Equation 48).To prove (Equation 51), it follows from the convexity of *f* that
$$f\left(\varphi_{0}\left(t\right)\right)\leq \frac{b-\varphi_{0}\left(t\right)}{b-a}f\left(a\right)+\frac{\varphi_{0}\left(t\right)-a}{b-a}f\left(b\right),\;t\in \left[a,b\right].$$ By integrating both sides of this inequality and using (Equation 48), we obtain the result.The proof is complete. □

**Remark** **6.**
*Assume that the conditions of Theorem 8 are satisfied.*

*(a) We obtain a method to refine both sides of inequalities (Equation 46) in Theorem 8.*

*(b) It is worth noting that further refinements of (Equation 46) can be obtained using the following observation: Define the functions $\varphi_{\lambda }:\left[a,b\right]\to \left[a,b\right]$
$\left(0\leq \lambda \leq 1\right)$ by*

$$\varphi_{\lambda }\left(t\right):=\left(1-\lambda \right)\varphi_{1}\left(t\right)+\lambda \varphi_{0}\left(t\right).$$


*Then it is easy to verify that for each $\lambda \in \left[0,1\right]$, the function $\varphi_{\lambda }$ is also increasing.*

*By the first inequality in (Equation 47),*

$$\int_{\left[a,x\right]}\varphi_{0}d\mu \leq \int_{\left[a,x\right]}\varphi_{\lambda }d\mu \leq \int_{\left[a,x\right]}\varphi_{1}d\mu ,\;x\in \left[a,b\right],\;\lambda \in \left[0,1\right],$$

*and by (Equation 48),*

$$\int_{\left[a,b\right]}\varphi_{\lambda }d\mu =\int_{\left[a,b\right]}td\mu \left(t\right),\;\lambda \in \left[0,1\right].$$


*Now, by applying Theorem 6 (b${}_{2}$), the convexity of f, and the fourth inequality in (Equation 50), we have that*

$$\int_{\left[a,b\right]}f\circ \varphi_{1}d\mu \leq \int_{\left[a,b\right]}f\circ \varphi_{\lambda }d\mu$$


$$\leq \left(1-\lambda \right)\int_{\left[a,b\right]}f\circ \varphi_{1}d\mu +\lambda \int_{\left[a,b\right]}f\circ \varphi_{0}d\mu \leq \int_{\left[a,b\right]}f\circ \varphi_{0}d\mu .$$


*Similarly, if we define the functions $\psi_{\lambda }:\left[a,b\right]\to \left[a,b\right]$
$\left(0\leq \lambda \leq 1\right)$ by*

$$\psi_{\lambda }\left(t\right):=\left(1-\lambda \right)\psi_{1}\left(t\right)+\lambda \psi_{0}\left(t\right),$$

*then*

$$\int_{\left[a,b\right]}f\circ \psi_{0}d\mu \leq \int_{\left[a,b\right]}f\circ \psi_{\lambda }d\mu$$


$$\leq \left(1-\lambda \right)\int_{\left[a,b\right]}f\circ \psi_{1}d\mu +\lambda \int_{\left[a,b\right]}f\circ \psi_{0}d\mu \leq \int_{\left[a,b\right]}f\circ \psi_{1}d\mu .$$


*(c) The results of many papers on the refinement of the Hermite-Hadamard inequality, whose proofs are based on different ideas, can be treated in a uniform way, taking into account the previous remark. See, for example, Theorem 1.1 in [13], Theorem 2.1 and Theorem 2.2 in [14], Theorem 3.1 and Theorem 3.4 in [15], Theorem 2.1, Theorem 2.7, and Theorem 2.8 in [16], and Theorem 1 in [17].*

*(d) A different approach to refining Fejér-, especially Hermite-Hadamard inequalities, can be found in [18].*


Now, we present general extensions of Fejér-, especially Hermite-Hadamard inequalities. Moreover, an efficient method is obtained for refining such inequalities.

**Theorem** **9.**
*Let $\left[a,b\right]\subset R$ with a<b, and let μ be a finite signed measure on $B_{\left[a,b\right]}$, such that (Equation 14) holds.*

*(a) The inequality*

$$f\left(\frac{a+b}{2}\right)\mu \left(\left[a,b\right]\right)\leq \int_{\left[a,b\right]}fd\mu$$

*holds for all convex functions f:a,b→R if and only if*

$$\int_{\left[a,x\right]}\left(x-t\right)d\mu \left(t\right)\geq 0,\;x\in \left[a,b\right].$$


*(b) The inequality*

$$\int_{\left[a,b\right]}fd\mu \leq \frac{f\left(a\right)+f\left(b\right)}{2}\mu \left(\left[a,b\right]\right)$$

*holds for all convex functions $f:\left[a,b\right]\to R$ if and only if (Equation 41) is satisfied.*

*(c) Assume μ is a measure and $\varphi_{1}$, $\varphi_{0}$
$\psi_{1}$, $\psi_{0}:\left[a,b\right]\to \left[a,b\right]$ are increasing functions, such that*

(54)
$$\int_{\left[a,x\right]}\varphi_{0}d\mu \leq \int_{\left[a,x\right]}\varphi_{1}d\mu \leq \int_{\left[a,x\right]}td\mu \left(t\right)\leq \int_{\left[a,x\right]}\psi_{1}d\mu \leq \int_{\left[a,x\right]}\psi_{0}d\mu ,\;x\in \left[a,b\right]$$

*and*

(55)
$$\int_{\left[a,b\right]}\varphi_{0}d\mu =\int_{\left[a,b\right]}\varphi_{1}d\mu =\int_{\left[a,b\right]}\psi_{1}d\mu =\int_{\left[a,b\right]}\psi_{0}d\mu =\frac{a+b}{2}\mu \left(\left[a,b\right]\right)$$

*are satisfied. Then, for all convex functions $f:\left[a,b\right]\to R$, we have*

(56)
$$f\left(\frac{a+b}{2}\right)\mu \left(\left[a,b\right]\right)$$


$$\leq \int_{\left[a,b\right]}f\circ \psi_{0}d\mu \leq \int_{\left[a,b\right]}f\circ \psi_{1}d\mu \leq \int_{\left[a,b\right]}fd\mu \leq \int_{\left[a,b\right]}f\circ \varphi_{1}d\mu \leq \int_{\left[a,b\right]}f\circ \varphi_{0}d\mu$$


(57)
$$\leq \frac{f\left(a\right)+f\left(b\right)}{2}\mu \left(\left[a,b\right]\right)+\left(f\left(\frac{a+b}{2}\right)-\frac{f\left(a\right)+f\left(b\right)}{2}\right)\mu \left(\left\{\frac{a+b}{2}\right\}\right).$$



**Proof.** Since the identity function on $\left[a,b\right]$ satisfies (Equation 15), it follows from (Equation 18) that $x_{\mu }=\frac{a+b}{2}$. By using this and the symmetry of the measure, Theorem 7 (a) and (b) imply (a) and (b), respectively.Inequality (Equation 56) comes from Theorem 7 (Equation 49).We need to prove (Equation 57).We introduce the increasing function $\varphi_{l}:\left[a,b\right]\to \left[a,b\right]$,
$$\varphi_{l}\left(t\right):=\left\{\begin{array}{c} a,\;a\leq t<\frac{a+b}{2}\\ \frac{a+b}{2},\;t=\frac{a+b}{2}\\ b,\;\frac{a+b}{2}<t\leq b \end{array}\right..$$By (Equation 18),
(58)$$\int_{\left[a,b\right]}td\mu \left(t\right)=\int_{\left[a,b\right]}\varphi_{l}d\mu =\frac{a+b}{2}\mu \left(\left[a,b\right]\right).$$Next, we show that
$$\int_{\left[a,x\right]}\varphi_{0}d\mu \geq \int_{\left[a,x\right]}\varphi_{l}d\mu ,\;x\in \left[a,b\right].$$This is obvious if $x\in \left[a,\frac{a+b}{2}\right]$. For $x=\frac{a+b}{2}$, suppose that, on the contrary,
(59)$$\int_{\left[a,\frac{a+b}{2}\right]}\varphi_{0}d\mu <\int_{\left[a,\frac{a+b}{2}\right]}\varphi_{l}d\mu =a\mu \left(\left[a,\frac{a+b}{2}\right]\right)+\frac{a+b}{2}\mu \left(\left\{\frac{a+b}{2}\right\}\right).$$ Then by using (Equation 58), (Equation 59) and $\varphi_{0}\left(t\right)\leq b$
$\left(t\in \left[a,b\right]\right)$, we obtain
$$\frac{a+b}{2}\mu \left(\left[a,b\right]\right)=\int_{\left[a,b\right]}\varphi_{0}d\mu =\int_{\left[a,\frac{a+b}{2}\right]}\varphi_{0}d\mu +\int_{\left[\frac{a+b}{2},b\right]}\varphi_{0}d\mu$$
$$<a\mu \left(\left[a,\frac{a+b}{2}\right]\right)+\frac{a+b}{2}\mu \left(\left\{\frac{a+b}{2}\right\}\right)+\int_{\left[\frac{a+b}{2},b\right]}\varphi_{0}d\mu$$
$$\leq a\mu \left(\left[a,\frac{a+b}{2}\right]\right)+\frac{a+b}{2}\mu \left(\left\{\frac{a+b}{2}\right\}\right)+b\mu \left(\left[\frac{a+b}{2},b\right]\right)$$
$$=\frac{a+b}{2}\mu \left(\left[a,b\right]\right)$$
which is a contradiction.Finally, assume that there exists $x\in \left[\frac{a+b}{2},b\right]$, such that
$$\int_{\left[a,x\right]}\varphi_{0}d\mu <\int_{\left[a,x\right]}\varphi_{l}d\mu$$
$$=a\mu \left(\left[a,\frac{a+b}{2}\right]\right)+\frac{a+b}{2}\mu \left(\left\{\frac{a+b}{2}\right\}\right)+b\mu \left(\left[\frac{a+b}{2},x\right]\right).$$ This implies by using (Equation 58) that
$$\frac{a+b}{2}\mu \left(\left[a,b\right]\right)=\int_{\left[a,b\right]}\varphi_{0}d\mu <a\mu \left(\left[a,\frac{a+b}{2}\right]\right)+\frac{a+b}{2}\mu \left(\left\{\frac{a+b}{2}\right\}\right)$$
$$+b\mu \left(\left[\frac{a+b}{2},x\right]\right)+\int_{\left[x,b\right]}\varphi_{0}d\mu .$$ Since $\varphi_{0}$ is increasing, it now follows from (Equation 14) that
$$\frac{a+b}{2}\mu \left(\left[a,b\right]\right)<\frac{a+b}{2}\mu \left(\left[a,b\right]\right)+\mu \left(\left[x,b\right]\right)\left(\varphi_{0}\left(b\right)-b\right)\leq \frac{a+b}{2}\mu \left(\left[a,b\right]\right)$$
which is also a contradiction.Now, Theorem 6 (b${}_{2}$) can be applied.The proof is complete. □

**Remark** **7.**
*Let $\left[a,b\right]\subset R$ with a<b, and let μ be a finite measure on $B_{\left[a,b\right]}$, such that (Equation 14) holds.*

*(a) Conditions (Equation 54) and (Equation 55) in the previous statement can be replaced by one of the following more easily checked conditions:*

*(i) The functions $\varphi_{1}$, $\varphi_{0}$
$\psi_{1}$, $\psi_{0}$ satisfy the symmetry property (Equation 15) and*

(60)
$$\int_{\left[a,x\right]}\varphi_{0}d\mu \leq \int_{\left[a,x\right]}\varphi_{1}d\mu \leq \int_{\left[a,x\right]}td\mu \left(t\right)\leq \int_{\left[a,x\right]}\psi_{1}d\mu \leq \int_{\left[a,x\right]}\psi_{0}d\mu ,\;x\in \left[a,\frac{a+b}{2}\right].$$


*(ii) The functions $\varphi_{1}$, $\varphi_{0}$
$\psi_{1}$, $\psi_{0}$ satisfy the symmetry property (Equation 15), and*

(61)
$$\varphi_{0}\left(t\right)\leq \varphi_{1}\left(t\right)\leq t\leq \psi_{1}\left(t\right)\leq \psi_{0}\left(t\right),\;t\in \left[a,\frac{a+b}{2}\right].$$


*Really, by Lemma 4 (a), (Equation 60) implies (Equation 54) and (Equation 55), and by Lemma 4 (b), (Equation 61) implies (Equation 60).*

*(b) We proved in Theorem 9 that*

(62)
$$f\left(\frac{a+b}{2}\right)\mu \left(\left[a,b\right]\right)\leq \int_{\left[a,b\right]}fd\mu \leq \frac{f\left(a\right)+f\left(b\right)}{2}\mu \left(\left[a,b\right]\right),$$

*moreover, (Equation 57) refines the right-hand side of (Equation 62).*

*The theorem also yields refinements of both the left-hand and right-hand inequalities in (Equation 62).*


Next, we highlight the following special case of the previous result, where we assume that μ is absolutely continuous with respect to the Lebesgue measure λ on $B_{\left[a,b\right]}$.

**Corollary** **4.**
*Let $\left[a,b\right]\subset R$ with a<b, and let $p:\left[a,b\right]\to R$ be a nonnegative and Lebesgue-integrable function for which*

(63)
$$p\left(t\right)=p\left(a+b-t\right),\;t\in \left[a,b\right].$$

*Let $\varphi_{1}$, $\varphi_{0}$
$\psi_{1}$, $\psi_{0}:\left[a,b\right]\to \left[a,b\right]$ be increasing functions, such that*

(64)
$$\int_{a}^{x}\varphi_{0}pd\lambda \leq \int_{a}^{x}\varphi_{1}pd\lambda \leq \int_{a}^{x}tp\left(t\right)d\lambda \left(t\right)\leq \int_{a}^{x}\psi_{1}pd\lambda \leq \int_{a}^{x}\psi_{0}pd\lambda ,\;x\in \left[a,b\right]$$

*and*

$$\int_{a}^{b}\varphi_{0}pd\lambda =\int_{a}^{b}\varphi_{1}pd\lambda =\int_{a}^{b}\psi_{1}pd\lambda =\int_{a}^{b}\psi_{0}pd\lambda =\frac{b^{2}-a^{2}}{2}$$

*are satisfied. If $f:\left[a,b\right]\to R$ is a convex function, then*

$$f\left(\frac{a+b}{2}\right)\int_{a}^{b}pd\lambda$$


$$\leq \int_{a}^{b}\left(f\circ \psi_{0}\right)pd\lambda \leq \int_{a}^{b}\left(f\circ \psi_{1}\right)pd\lambda \leq \int_{a}^{b}fpd\lambda \leq \int_{a}^{b}\left(f\circ \varphi_{1}\right)pd\lambda \leq \int_{a}^{b}\left(f\circ \varphi_{0}\right)pd\lambda$$


$$\leq \frac{f\left(a\right)+f\left(b\right)}{2}\int_{a}^{b}pd\lambda .$$



**Proof.** By (Equation 63), the measure μ defined on $B_{\left[a,b\right]}$ by
$$\mu \left(A\right):=\int_{A}pd\lambda$$
satisfies (Equation 14) and, thus, Theorem 9 (c) can be applied.The proof is complete. □

**Remark** **8.**
*Assume the conditions of Corollary 4 are satisfied.*

*(a) Similar to Remark 7 (a), if the functions $\varphi_{1}$, $\varphi_{0}$
$\psi_{1}$, $\psi_{0}$ satisfy the symmetry property (Equation 15), then any of conditions (Equation 60) and (Equation 61) may be used instead of (Equation 64).*

*(b) It can be seen that Fejér’s inequality*

(65)
$$f\left(\frac{a+b}{2}\right)\int_{a}^{b}pd\lambda \leq \int_{a}^{b}fpd\lambda \leq \frac{f\left(a\right)+f\left(b\right)}{2}\int_{a}^{b}pd\lambda$$

*and especially the Hermite-Hadamard inequality*

$$f\left(\frac{a+b}{2}\right)\leq \int_{a}^{b}fd\lambda \leq \frac{f\left(a\right)+f\left(b\right)}{2}$$

*are very special cases of Theorem 9.*

*(c) In Corollary 4, we also obtained a method (see Remark 7 (c)) for refining both the left-hand side and the right-hand side inequality of (Equation 65).*


## 5. Application to *f*-Divergences

The following notion was introduced by Csiszár in [19,20].

**Definition** **4.**
*Let $f:\left[0,\infty \right]\to \left[0,\infty \right]$ be a convex function, and let $p:=\left(p_{1},\ldots ,p_{n}\right)$ and $q:=\left(q_{1},\ldots ,q_{n}\right)$ be positive probability distributions. The f-functional divergence is*

$$I_{f}\left(p,q\right):=\sum_{i=1}^{n}q_{i}f\left(\frac{p_{i}}{q_{i}}\right).$$



It is possible to use nonnegative probability distributions in the *f*-functional divergence, by defining
$$f\left(0\right):=\lim_{t\to 0+}f\left(t\right);\;0f\left(\frac{0}{0}\right):=0;\;0f\left(\frac{a}{0}\right):=\lim_{t\to 0+}tf\left(\frac{a}{t}\right),\;a>0.$$

The basic inequality (which comes from the discrete Jensen inequality)
(66)$$I_{f}\left(p,q\right)\geq f\left(1\right)$$
is one of the key properties of *f*-divergences.

The refinement of inequality (Equation 66) is the subject of several papers (for a non-exhaustive list, see [21] and references therein, and papers [22,23,24,25]). In the following statement, we present a necessary and sufficient condition for the inequality
$$I_{f}\left(p,q\right)\geq I_{f}\left(u,v\right)$$
to be satisfied; thus, we obtain a necessary and sufficient condition for refining inequality (Equation 66) by another *f*-divergence.

**Theorem** **10.**
*Let $X:=\left\{1,\ldots ,n\right\}$ for some n≥1, and let $Y:=\left\{1,\ldots ,m\right\}$ for some m≥1. Let $p:=\left(p_{1},\ldots ,p_{n}\right)$, $q:=\left(q_{1},\ldots ,q_{n}\right)$, $u:=\left(u_{1},\ldots ,u_{m}\right)$ and $v:=\left(v_{1},\ldots ,v_{m}\right)$ be positive probability distributions. Let $c_{1}>c_{2}>\ldots >c_{k}$ be the different elements of ${\left(\frac{p_{i}}{q_{i}}\right)}_{i=1}^{n}$ and ${\left(\frac{u_{j}}{v_{j}}\right)}_{j=1}^{m}$ in decreasing order $\left(1\leq k\leq m+n\right)$. For every convex function $f:\left[0,\infty \right]\to \left[0,\infty \right]$ inequality*

(67)
$$\sum_{i=1}^{n}q_{i}f\left(\frac{p_{i}}{q_{i}}\right)=I_{f}\left(p,q\right)\geq I_{f}\left(u,v\right)=\sum_{j=1}^{m}v_{j}f\left(\frac{u_{j}}{v_{j}}\right)$$

*holds if and only if*

$$\sum_{\left\{j\in Y|\frac{u_{j}}{v_{j}}\geq c_{l}\right\}}u_{j}-\sum_{\left\{i\in X|\frac{p_{i}}{q_{i}}\geq c_{l}\right\}}p_{i}$$


(68)
$$\leq c_{l}\left(\sum_{\left\{j\in Y|\frac{u_{j}}{v_{j}}\geq c_{l}\right\}}v_{j}-\sum_{\left\{i\in X|\frac{p_{i}}{q_{i}}\geq c_{l}\right\}}q_{i}\right),\;l=1,\ldots ,k.$$



**Proof.** Let $\left[a,b\right]\subset \left[0,\infty \right]$, such that $a\leq c_{k}<c_{1}\leq b$.Define the probability measures μ and ν on $B_{\left[a,b\right]}$ by
$$\nu :=\sum_{i=1}^{n}q_{i}\varepsilon_{p_{i}/q_{i}}\;\mathrm{and}\;\mu :=\sum_{j=1}^{m}v_{j}\varepsilon_{u_{j}/v_{j}},$$
and let φ, $\psi :\left[a,b\right]\to \left[0,\infty \right]$, $\varphi \left(t\right)=\psi \left(t\right):=t$.Then $\varphi \in L\left(\left[a,b\right],\mu \right)$, $\psi \in L\left(\left[a,b\right],\nu \right)$, $f\circ \varphi \in L\left(\left[a,b\right],\mu \right)$, $f\circ \psi \in L\left(\left[a,b\right],\nu \right)$ and
$$I_{f}\left(p,q\right)=\int_{\left[a,b\right]}\psi d\nu \;\mathrm{and}\;I_{f}\left(u,v\right)=\int_{\left[a,b\right]}\varphi d\mu .$$By Theorem 5 (c), inequality (Equation 67) holds if and only if it is satisfied in the following special cases: function *f* is $p_{\left[0,\infty \right],x}$
$\left(x\in \left[0,\infty \right]\right)$. This means that inequality (Equation 67) holds if and only if
$$\sum_{\left\{j\in Y|\frac{u_{j}}{v_{j}}\geq x\right\}}v_{j}\left(\frac{u_{j}}{v_{j}}-x\right)\leq \sum_{\left\{i\in X|\frac{p_{i}}{q_{i}}\geq x\right\}}q_{i}\left(\frac{p_{i}}{q_{i}}-x\right),\;x\in \left[0,\infty \right],$$
or, equivalently,
$$\sum_{\left\{j\in Y|\frac{u_{j}}{v_{j}}\geq x\right\}}u_{j}-\sum_{\left\{i\in X|\frac{p_{i}}{q_{i}}\geq x\right\}}p_{i}$$
(69)$$\leq x\left(\sum_{\left\{j\in Y|\frac{u_{j}}{v_{j}}\geq x\right\}}v_{j}-\sum_{\left\{i\in X|\frac{p_{i}}{q_{i}}\geq x\right\}}q_{i}\right),\;x\in \left[0,\infty \right].$$It follows that it is enough to prove the equivalence of (Equation 68) and (Equation 69).It is obvious that (Equation 69) implies (Equation 68).Conversely, assume (Equation 68) is satisfied, and let $c_{l+1}<x\leq c_{l}$ for some 1≤l<k.Then
$$\sum_{\left\{j\in Y|\frac{u_{j}}{v_{j}}\geq c_{l}\right\}}u_{j}-\sum_{\left\{i\in X|\frac{p_{i}}{q_{i}}\geq c_{l}\right\}}p_{i}+\sum_{\left\{j\in Y|\frac{u_{j}}{v_{j}}=c_{l+1}\right\}}u_{j}-\sum_{\left\{i\in X|\frac{p_{i}}{q_{i}}=c_{l+1}\right\}}p_{i}$$
$$=\sum_{\left\{j\in Y|\frac{u_{j}}{v_{j}}\geq c_{l+1}\right\}}u_{j}-\sum_{\left\{i\in X|\frac{p_{i}}{q_{i}}\geq c_{l+1}\right\}}p_{i}$$
$$\leq c_{l+1}\left(\sum_{\left\{j\in Y|\frac{u_{j}}{v_{j}}\geq c_{l+1}\right\}}v_{j}-\sum_{\left\{i\in X|\frac{p_{i}}{q_{i}}\geq c_{l+1}\right\}}q_{i}\right)$$
$$=c_{l+1}\left(\sum_{\left\{j\in Y|\frac{u_{j}}{v_{j}}\geq c_{l}\right\}}v_{j}-\sum_{\left\{i\in X|\frac{p_{i}}{q_{i}}\geq c_{l}\right\}}q_{i}\right)$$
$$+c_{l+1}\left(\sum_{\left\{j\in Y|\frac{u_{j}}{v_{j}}=c_{l+1}\right\}}v_{j}-\sum_{\left\{i\in X|\frac{p_{i}}{q_{i}}=c_{l+1}\right\}}q_{i}\right)$$
$$=c_{l+1}\left(\sum_{\left\{j\in Y|\frac{u_{j}}{v_{j}}\geq c_{l}\right\}}v_{j}-\sum_{\left\{i\in X|\frac{p_{i}}{q_{i}}\geq c_{l}\right\}}q_{i}\right)$$
$$+\sum_{\left\{j\in Y|\frac{u_{j}}{v_{j}}=c_{l+1}\right\}}u_{j}-\sum_{\left\{i\in X|\frac{p_{i}}{q_{i}}=c_{l+1}\right\}}p_{i},$$
and, therefore,
(70)$$\sum_{\left\{j\in Y|\frac{u_{j}}{v_{j}}\geq c_{l}\right\}}u_{j}-\sum_{\left\{i\in X|\frac{p_{i}}{q_{i}}\geq c_{l}\right\}}p_{i}\leq c_{l+1}\left(\sum_{\left\{j\in Y|\frac{u_{j}}{v_{j}}\geq c_{l}\right\}}v_{j}-\sum_{\left\{i\in X|\frac{p_{i}}{q_{i}}\geq c_{l}\right\}}q_{i}\right).$$It now follows from (Equation 68) and (Equation 70) that
$$\sum_{\left\{j\in Y|\frac{u_{j}}{v_{j}}\geq x\right\}}u_{j}-\sum_{\left\{i\in X|\frac{p_{i}}{q_{i}}\geq x\right\}}p_{i}=\sum_{\left\{j\in Y|\frac{u_{j}}{v_{j}}\geq c_{l}\right\}}u_{j}-\sum_{\left\{i\in X|\frac{p_{i}}{q_{i}}\geq c_{l}\right\}}p_{i}$$
$$\leq c_{l}\left(\sum_{\left\{j\in Y|\frac{u_{j}}{v_{j}}\geq c_{l}\right\}}v_{j}-\sum_{\left\{i\in X|\frac{p_{i}}{q_{i}}\geq c_{l}\right\}}q_{i}\right)$$
$$=c_{l}\left(\sum_{\left\{j\in Y|\frac{u_{j}}{v_{j}}\geq x\right\}}v_{j}-\sum_{\left\{i\in X|\frac{p_{i}}{q_{i}}\geq x\right\}}q_{i}\right)$$
and
$$\sum_{\left\{j\in Y|\frac{u_{j}}{v_{j}}\geq x\right\}}u_{j}-\sum_{\left\{i\in X|\frac{p_{i}}{q_{i}}\geq x\right\}}p_{i}\leq c_{l+1}\left(\sum_{\left\{j\in Y|\frac{u_{j}}{v_{j}}\geq x\right\}}v_{j}-\sum_{\left\{i\in X|\frac{p_{i}}{q_{i}}\geq x\right\}}q_{i}\right)$$
and these imply (Equation 69).The proof is complete. □

**Remark** **9.**
*We emphasize that the test for the inequalities in (Equation 68) is finite and easily verifiable.*


## 6. Conclusions

In this paper, we studied majorization-type integral inequalities by using finite signed measures. Necessary and sufficient conditions were given for the inequalities under consideration to be satisfied. In order to achieve this goal, we generalized the statement on the approximation of convex functions defined on compact intervals by piecewise linear convex functions to arbitrary intervals. This in itself is an interesting and useful result. To apply these results, we first dealt with Hermite-Hadamard–Fejér-type inequalities and their refinements. Along with new results, we obtained unified and simple proofs of some classical statements. Finally, we obtained a general method to refine both sides of Hermite-Hadamard-Fejér-type inequalities. The results of many papers on the refinement of the Hermite-Hadamard inequality, where proofs are based on different ideas, can be treated in a uniform way by this method. The results obtained and the methods used can be useful in many areas. Finally, we established a necessary and sufficient condition for when a fundamental inequality of *f*-divergences can be refined by another *f*-divergence.

## Data Availability

Not applicable.

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
