# Peer review of "Uniform Treatment of Integral Majorization Inequalities with Applications to Hermite-Hadamard-Fejér-Type Inequalities and f-Divergences"

_entropy, 2023, doi:10.3390/e25060954_

Round 1

Reviewer 1 Report

See pdf file

Reviewer 2 Report

I recommend this paper for publication after minor revision. Please see the attached file.

Minor editing of English language is required. 

Reviewer 3 Report

The author proposes a uniform treatment for several majorization inequalities, including the classical Hermite-Hatamard inequality and its several generalizations. According to my knowledge, the results are new. The proofs appear to be correct. Moreover, the exposition is very clear.

I have only one small remark: lines 279-280: it is better to keep all notational conventions at the beginning of the paper, so I suggest moving the definition of the set of positive integers and the definition of the interior to Section 2.
